# Effectiveness of a Digital Game-Based Physical Activity Program (AI-FIT) on Health-Related Physical Fitness in Elementary School Children

**DOI:** 10.3390/healthcare13111327

**Published:** 2025-06-03

**Authors:** Se-Won Park, Dong-Ha Lim, Je-Hyun Kim, Sung-Hun Kim, Yeon-Oh Han

**Affiliations:** 1Department of Physical Education, Korea National University of Education, Cheongju 28173, Republic of Korea; parksewon@knue.ac.kr; 2Research Institute, AIRPASS Co., Ltd., Hanam-si 12982, Gyeonggi-do, Republic of Korea; limjack56@airpass.co.kr (D.-H.L.); didino35@airpass.co.kr (J.-H.K.); hun1564@airpass.co.kr (S.-H.K.); 3HealthPro LAB, Yeouido, Yeongdeungpo-gu, Seoul 07325, Republic of Korea

**Keywords:** active video games, personalized exercise, gamification, child health promotion, physical activity intervention, AI in education

## Abstract

Objectives: This study empirically examined the effects of a digital game-based physical activity program (AI-FIT) on elementary school children’s health-related physical fitness while exploring the role of educational technology (EdTech) as a practical solution to post-pandemic physical inactivity. Methods: This study was conducted over a 12-week period, from September to December 2024, with 40 students (grades 4 to 6) from an elementary school located in a mid-sized city in South Korea. The participants had a mean age of 10.8 years (SD = 0.69). The experimental group (n = 20) participated in the AI-FIT program, while the control group (n = 20) received regular physical education classes. To ensure baseline equivalence between the groups, propensity score matching was employed. Health-related physical fitness was assessed through pre- and post-tests, including flexibility, muscular endurance, cardiorespiratory endurance, explosive power, and Physical Activity Promotion System (PAPS) grade. Analyses were conducted using both dependent (paired) and independent *t*-tests, along with effect size calculations (Cohen’s *d*), to examine within-group and between-group differences. In addition, gender-based subgroup analyses were performed to examine potential differences in responsiveness to the intervention. Intervention: Results indicated that the AI-FIT program had a large effect on flexibility (*d* = 0.90) and muscular endurance (*d* = 0.80) and a moderate-to-large effect on PAPS grade (*d* = 0.69). In contrast, no statistically significant improvements were observed in cardiorespiratory endurance or explosive power. Conclusions: These findings suggest that a digital program integrating AI-based personalized exercise prescriptions and gamification elements can effectively enhance the health-related fitness of elementary students. Moreover, this study supports the educational efficacy of EdTech-based interventions in physical education settings.

## 1. Introduction

In recent years, the global decline in physical activity and physical fitness among children and adolescents has emerged as a critical educational concern, with potential implications not only for individual health but also for long-term socio-economic productivity [1,2]. This concern has been further exacerbated by the COVID-19 pandemic, which disrupted physical education (PE) classes and prolonged remote learning, thereby limiting physical activity opportunities during crucial developmental years and accelerating declines in muscular strength, cardiorespiratory endurance, and flexibility.

According to the World Health Organization (WHO), children and adolescents aged 5–17 years should engage in an average of at least 60 min of moderate-to-vigorous intensity physical activity daily, primarily aerobic, to improve their cardiorespiratory and muscular fitness, bone health, and metabolic function. However, global data indicate that over 80% of this population fails to meet the recommended levels [3].

In parallel, rapid advancements in cutting-edge digital technologies—such as AI, AR, VR, and MR—have significantly reshaped all sectors of society, including education. These technologies are now considered core drivers in the so-called VUCA era, characterized by volatility, uncertainty, complexity, and ambiguity [4,5]. While they offer immense potential to drive innovation in education, they also pose risks by promoting sedentary behavior among children, thereby reducing physical activity and negatively affecting health outcomes [6]. Furthermore, excessive exposure to AI-driven media and screen-based environments has been associated with reduced attention span, impaired sleep patterns, and delayed social–emotional development in children [7,8]. These negative effects underscore the importance of integrating technology in ways that actively promote movement, interaction, and physical engagement.

To address these challenges, there is a growing demand for strategic approaches that move beyond passive use of technology and instead meaningfully integrate it into educational contexts to enhance children’s physical activity and fitness. Recently, educational technology (EdTech) combining digital game-based learning, immersive technologies (e.g., VR and AR), and AI has gained significant attention. These innovations have the potential to enhance learner engagement and autonomy while enabling personalized physical activity interventions, thus improving both the effectiveness and equity of physical education.

Several empirical studies support the potential of these tools. For example, Huang et al. [9] demonstrated that virtual reality (VR) environments grounded in constructivist principles significantly improve learners’ motivation and perceived effectiveness. Klopfer and Sheldon [10] found that student-authored augmented reality (AR) games enhanced science learning by promoting interactive and embodied experiences. In a more applied context, Deutsch et al. [11] reported that low-cost gaming consoles like the Nintendo Wii can support physical rehabilitation in adolescents with cerebral palsy, illustrating the broader applicability of such tools to physical activity interventions.

Since Klaus Schwab’s discourse on the Fourth Industrial Revolution, the education sector has actively pursued the integration of EdTech. This trend accelerated during the COVID-19 pandemic, which spurred the adoption of hyperconnected systems—such as the Internet of Things (IoT), cloud platforms, and wireless networks—thereby facilitating remote learning and digital transformation [12]. However, in the context of physical education (PE)—particularly a practice-oriented subject—this rapid transition had mixed effects. While some digital tools helped sustain student interaction, many PE teachers reported difficulties in delivering embodied, movement-based instruction remotely, which led to decreased student motivation and a temporary erosion of PE’s identity as a physically engaging subject [13,14,15].

Nevertheless, when effectively implemented, EdTech has shown promise in enhancing learning engagement, student participation, and tangible improvements in physical fitness. VR-based instruction can positively influence learner immersion and motor skill development. For example, Steuer [16] defined the concept of telepresence as the psychological state of “being there” in a mediated environment, laying the theoretical foundation for immersive learning experiences. Finkenberg and Mohnsen [17] highlighted the pedagogical potential of VR in physical education, emphasizing its ability to increase learner motivation, simulate real-world movement tasks, and support skill acquisition through engaging scenarios. AR, in particular, has garnered attention for its ability to augment interactive information while maintaining real-world context, making it highly applicable in both indoor and outdoor settings [18]. Moreover, digital game-based physical activity has been linked to increased learner participation and sustained engagement, ultimately supporting improvements in physical fitness [19].

Nonetheless, the majority of prior research has focused on theoretical disciplines, with relatively limited application of immersive or AI-driven technologies in physical education [13,14,20]. More recently, AI-based systems have demonstrated the ability to analyze students’ activity data, fitness levels, and behavioral patterns, enabling individualized feedback and customized exercise prescriptions [21,22]. Such technologies also support the development of automated feedback systems for analyzing student engagement and enhancing instructional strategies in PE classes [23]. These advancements represent a shift from viewing technology as a peripheral tool to integrating it as a core component of instructional practice, with the potential to fundamentally improve the quality of physical education [24].

Accordingly, the present study aims to empirically examine the effects of *AI-FIT*, a digital game-based physical activity program, on the health-related physical fitness of elementary school students over a 12-week period. Specifically, this study investigates whether the program yields significant improvements in muscular endurance, cardiorespiratory endurance, flexibility, explosive power, and PAPS (Physical Activity Promotion System) grade. It is hypothesized that students participating in the AI-FIT program will demonstrate significant improvements across these components compared to those in the control group.

The central research question is the following: Does the AI-FIT program produce measurable changes in elementary students’ health-related fitness—namely, flexibility, muscular endurance, cardiorespiratory endurance, explosive power, and PAPS grade?

## 2. Materials and Methods

### 2.1. Participants and Research Design

This study employed a quasi-experimental design to verify the effects of a digital game-based physical activity program (AI-FIT) on the health-related physical fitness of elementary school students. The participants were 40 students in grades 4 to 6 (mean age = 10.8 years, SD = 0.69) from an elementary school located in City C, a mid-sized city in South Korea. The experimental group (n = 20) participated in the AI-FIT program, while the control group (n = 20) continued their regular school physical education classes without participating in any after-school programs. These classes included general physical activities aligned with the national curriculum, such as warm-up exercises, basic motor skills, and cooperative games. In contrast, the experimental group participated in the AI-FIT program as an additional after-school activity.

To ensure baseline equivalence between the groups, propensity scores were calculated using logistic regression based on covariates including gender, grade level, body height, body weight, and pre-test scores on the Physical Activity Promotion System (PAPS). A 1:1 nearest-neighbor matching procedure was then applied. As a result of the matching, no statistically significant differences were observed between the experimental and control groups in baseline characteristics (gender, grade, body height, and body weight; *p* > 0.05), indicating that group homogeneity was secured.

The general characteristics of the participants are summarized in Table 1, and their distributions in body height and body weight between groups are visually presented in Figure 1. This study was approved by the Institutional Review Board (IRB) of Korea National University of Education (Approval No.: KNUE-202410-SB-0601-01).

All physical fitness assessments were conducted by certified personnel from the National Fitness Award Center in accordance with standardized procedures. Each participant was measured by the same evaluator before and after the intervention to ensure inter-rater consistency. Although multiple trained evaluators were involved, each subject’s test was conducted by a single consistent tester, eliminating the need for inter-observer reliability analysis.

Additional participant characteristics are presented as follows. Among the total 40 participants, 16 were male (40%) and 24 were female (60%). By grade level, 14 students were in grade 4 (35%), 20 in grade 5 (50%), and 6 in grade 6 (15%). The average body height was 148.2 cm (±6.02), and the average body weight was 53.4 kg (±14.86).

Statistical analyses confirmed no significant differences between the experimental and control groups in terms of gender, grade, body height, and body weight (*p* > 0.05), both before and after the application of propensity score matching. These findings support the appropriateness of group equivalence.

Although the total sample size (n = 40) was relatively small, this study ensured strict experimental control, applied propensity score matching to secure baseline homogeneity, and conducted effect size analyses (Cohen’s *d*). Accordingly, this study may be interpreted as a valid and empirically reliable comparative investigation [25,26]. This sample size is also consistent with those commonly used in pilot efficacy trials or small-scale early-phase health intervention studies involving digital physical activity programs.

### 2.2. Intervention Program: AI-FIT

The AI-FIT program utilized in this study is a digital physical activity system based on a mixed reality (MR) kiosk interface, designed to provide AI-driven, personalized exercise prescriptions tailored to the individual physical fitness levels of elementary school students. The program requires one or two students at a time to perform targeted movements for approximately 20 s by following a virtual interface projected onto the floor. Based on the students’ performance data, the AI algorithm dynamically adjusts the intensity and movement path in real time.

A key feature of AI-FIT is its capacity to offer individualized exercise tasks rather than standardized activities for all participants. Exercise prescriptions are customized according to students’ grade level, gender, and baseline fitness level. The system incorporates gamification elements to enhance engagement and motivation, and it primarily consists of bodyweight exercises, making it suitable for both school and home use.

The AI-FIT system includes 16 types of body weight-based exercise programs, such as arm walking, planks, box runs, and speed steps. These exercises are designed to stimulate a range of fitness components, including muscular endurance, flexibility, cardiorespiratory endurance, and explosive power. The program was implemented three times per week for 40 min per session over a 12-week period. The exercises were structured and categorized according to the targeted fitness components, as shown in Table 2.

The intervention was conducted over a 12-week period, from 4 September to 29 November 2024, targeting students in grades 4 to 6 at an elementary school located in City C, Chungcheongnam-do, South Korea. The experimental group participated in the AI-FIT program three times per week—on Mondays, Wednesdays, and Fridays—from 10:30 a.m. to 11:10 a.m. for 40 min per session. The intervention was implemented during regular physical education class hours using mixed reality-based digital exercise equipment installed in the school gymnasium.

Each 40-min AI-FIT session consisted of three phases: a 5-min warm-up, a 30-min main activity, and a 5-min cool-down. The warm-up included dynamic stretching and mobility exercises (e.g., jumping jacks, arm swings, and trunk rotations), while the cool-down focused on static stretching and deep breathing. The main activity involved digital game-based exercises targeting muscular endurance, cardiorespiratory endurance, flexibility, and explosive power, with real-time feedback and adaptive difficulty. The intervention commenced only after informed consent was obtained from participants and guardians and baseline fitness assessments were completed.

Exercise intensity was not adjusted in real time but rather prescribed progressively based on each participant’s previous performance data. The AI-FIT system analyzed indicators such as completion accuracy, response speed, and success rate after each session. Based on this analysis, it prescribed personalized digital exercise tasks for subsequent sessions, targeting moderate-to-vigorous physical activity (MVPA) levels (approximately 4–6 METs or RPE 12–14), aligned with each student’s developmental level and fitness status.

As the AI-FIT intervention was conducted as an afterschool program, the control group did not participate in any structured physical activity during the same time period. Both groups attended the same regular PE classes during the school day.

### 2.3. Measurement Instruments and Assessment Variables

Health-related physical fitness was assessed using the following standardized methods. Flexibility was measured using the sit-and-reach test, and muscular endurance was assessed by the number of sit-ups performed in one minute following standardized fitness testing protocols. Explosive power was measured through the standing long jump, and cardiorespiratory endurance was evaluated using a standardized shuttle run test, known as the Physical Endurance Index (PEI). In this test, participants were required to run back and forth across a fixed distance within a designated time limit, and the number of completed laps was recorded to determine their endurance level.

Although explosive power is not traditionally considered a core component of health-related physical fitness, it is included in the national PAPS framework. Therefore, it was measured in this study to reflect the real-world standards of school-based physical fitness assessment in Korea.

These measurement procedures are consistent with the protocols used in large-scale national assessments and intervention studies targeting Korean elementary students [27,28,29]. For instance, muscular strength was assessed using handgrip dynamometry, explosive power through the standing long jump with two trials, and shuttle run tests were conducted over 15 m intervals with standardized audio signals. Flexibility was measured using the sit-and-reach test, with two attempts averaged [29]. These methods are recognized for their reliability and feasibility in the school setting.

For a comprehensive assessment of health-related fitness, this study employed the Physical Activity Promotion System (PAPS) grade, a nationally standardized physical fitness evaluation tool used in Korean school settings. Administered under the supervision of the Ministry of Education, PAPS assesses five components—cardiorespiratory endurance, muscular endurance, flexibility, and explosive power—and provides a composite diagnostic score for students’ physical fitness. The instrument has been regarded as both reliable and valid [27] and has been widely applied to objectively evaluate the effectiveness of school-based physical activity programs [30].

PAPS classifies students into five performance levels, ranging from Grade 1 (Excellent) to Grade 5 (Needs Improvement), based on composite scores across the five fitness domains. In this study, the PAPS grade was treated as an interval variable to enable calculation of means and standard deviations for statistical analysis.

All assessments were conducted under consistent pre- and post-intervention conditions. Trained evaluators, who had completed prior instruction in the assessment protocols, were responsible for all measurements to ensure reliability.

### 2.4. Data Analysis

All data were analyzed using IBM SPSS Statistics version 28.0. Within-group pre- and post-intervention differences for both the experimental and control groups were analyzed using paired *t*-tests. Between-group differences in change scores were examined using independent *t*-tests.

Effect sizes for each physical fitness variable were calculated using Cohen’s *d*. Effect sizes were interpreted as small (≥0.2), medium (≥0.5), and large (≥0.8). In addition, a subgroup analysis was conducted within the experimental group to examine differences in program effects by gender; the control group was excluded from this analysis.

The level of statistical significance was set at *p* < 0.05 (*) and *p* < 0.01 (**). Furthermore, to confirm baseline equivalence between groups after propensity score matching, independent *t*-tests and chi-square (χ^2^) tests were conducted on the matched covariates.

## 3. Results

The results indicated that the experimental group showed significant improvements in multiple fitness indicators, including flexibility, muscular endurance, and overall PAPS grade. In contrast, the control group demonstrated little to no improvement in most variables and, in some cases, even a decline in performance. The key findings are summarized as follows.

### 3.1. Changes in Flexibility

Flexibility was assessed using the sit-and-reach test. In the experimental group, the mean flexibility score significantly increased from 6.10 cm (±6.33) at pre-test to 8.65 cm (±5.89) at post-test (*t* = –4.03, *p* = 0.001), with a mean change of +2.55 cm (±2.83). The effect size (Cohen’s *d*) was 0.90, indicating a large effect.

In contrast, the control group showed a slight decrease in flexibility from 7.95 cm (±7.83) to 7.13 cm (±8.82), which was not statistically significant (*t* = 0.70, *p* = 0.491). The effect size was –0.16, indicating a negligible effect.

As shown in Table 3 and Figure 2, these results suggest that digital game-based physical activity can be particularly effective in improving flexibility among children.

In the gender-specific subgroup analysis within the experimental group, both male and female students demonstrated statistically significant improvements in flexibility. Male students improved from a pre-test mean of 8.00 cm (±5.21) to a post-test mean of 10.67 cm (±4.97) (*p* = 0.019), while female students improved from 7.91 cm (±5.81) to 10.36 cm (±6.38) (*p* = 0.003). As shown in Table 4 and Figure 3, these findings suggest that the AI-FIT program consistently had a positive effect on enhancing flexibility in children, regardless of gender.

### 3.2. Changes in Muscular Endurance

Muscular endurance was assessed by the number of sit-ups performed within one minute. In the experimental group, the mean number of sit-ups significantly increased from 22.15 (±11.95) at pre-test to 38.00 (±26.49) at post-test (*t* = –3.58, *p* = 0.002), with a mean change of +15.85 (±19.78). The effect size (Cohen’s *d*) was 0.80, indicating a large effect.

In contrast, the control group showed an increase from 46.32 (±36.03) to 53.84 (±60.81), but the difference was not statistically significant (*t* = –0.54, *p* = 0.597), and the effect size was only 0.12, indicating a negligible effect.

As shown in Table 5 and Figure 4, these findings suggest that the AI-FIT program had a positive effect on improving muscular endurance in elementary school students.

In the gender-specific subgroup analysis within the experimental group, changes in muscular endurance were not statistically significant for either male or female students. Male students showed a decrease from a pre-test mean of 35.11 repetitions (±16.62) to a post-test mean of 30.11 repetitions (±15.35), which was not statistically significant (*p* = 0.083). Female students also demonstrated a decline, from 30.36 repetitions (±13.32) to 26.36 repetitions (±13.63), with the result approaching but not reaching statistical significance (*p* = 0.058).

As shown in Table 6 and Figure 5, these findings suggest that while muscular endurance significantly improved at the overall experimental group level, statistically significant changes were not observed within gender-specific subgroups.

### 3.3. Changes in Overall Physical Fitness (PAPS Grade)

Although the PAPS grade is an ordinal scale ranging from 1 to 5, it was treated as an interval scale in this study to allow for the calculation of means and standard deviations. The PAPS (Physical Activity Promotion System) consists of five levels, where Grade 1 indicates excellent physical fitness and Grade 5 indicates poor fitness. Thus, a decrease in the PAPS grade represents an improvement in overall health-related fitness.

In the experimental group, the mean PAPS grade significantly improved from 2.90 (±0.64) at pre-test to 2.75 (±0.53) at post-test (*t* = 3.11, *p* = 0.006), with an effect size (Cohen’s *d*) of 0.69, indicating a moderate-to-large effect. In contrast, the control group showed no significant change, with the mean grade slightly increasing from 2.80 (±0.65) to 2.85 (±0.70) (*t* = –0.41, *p* = 0.684), and a negligible effect size of –0.07.

Additionally, a nonparametric Wilcoxon signed-rank test was conducted to further validate the findings. The results revealed a statistically significant improvement in the experimental group (*Z* = –3.720, *p* < 0.001), as shown in Table 7 and Figure 6, reinforcing the reliability of the observed enhancement in overall physical fitness as measured by the PAPS grade.

In the gender-specific subgroup analysis within the experimental group, both male and female students demonstrated statistically significant improvements in PAPS grade. Male students improved from a pre-test mean of 3.00 (±0.87) to a post-test mean of 2.67 (±0.87) (*p* = 0.013), while female students showed an even greater improvement, from 2.82 (±0.70) to 2.73 (±0.74) (*p* = 0.001). As shown in Table 8 and Figure 7, these results suggest that the AI-FIT program had a positive effect on enhancing overall physical fitness in children, regardless of gender.

### 3.4. Changes in Cardiorespiratory Endurance

Cardiorespiratory endurance was assessed using a standardized shuttle run test (Physical Endurance Index, PEI). In the experimental group, the mean score slightly decreased from 65.77 (±21.98) at pre-test to 62.95 (±18.88) at post-test, but this change was not statistically significant (*t* = 0.52, *p* = 0.607). The mean change was –2.82 (±23.54), and the effect size (Cohen’s *d*) was –0.12, indicating a negligible effect.

In contrast, the control group showed a statistically significant decrease, with the mean score dropping from 77.54 (±17.28) to 68.50 (±17.22) (*t* = 3.36, *p* = 0.002). The mean change was –9.04 (±17.20), and the effect size was –0.56, indicating a moderate decline.

As shown in Table 9 and Figure 8, these findings suggest that while the AI-FIT program did not lead to significant improvements in cardiorespiratory endurance, it may have played a role in mitigating the decline observed in the control group.

In the gender-specific subgroup analysis within the experimental group, male students showed a slight, non-significant increase in cardiorespiratory endurance, from a pre-test mean of 30.33 repetitions (±11.42) to a post-test mean of 31.11 repetitions (±13.18) (*p* = 0.683). Female students also demonstrated an upward trend, increasing from 28.45 repetitions (±15.13) to 33.00 repetitions (±13.98), but the change did not reach statistical significance (*p* = 0.091). As shown in Table 10 and Figure 9, these results indicate that no clear improvements in cardiorespiratory endurance were observed in either gender subgroup.

### 3.5. Changes in Explosive Power

Explosive power was evaluated using the standing long jump test. In the experimental group, the mean jump distance slightly decreased from 124.60 cm (±25.30) at pre-test to 123.80 cm (±47.08) at post-test, but this change was not statistically significant (*t* = 0.07, *p* = 0.941). The mean change was –0.80 cm (±47.76), and the effect size (Cohen’s *d*) was –0.02, indicating virtually no effect.

Similarly, the control group showed a decline from a pre-test mean of 118.85 cm (±34.10) to a post-test mean of 108.25 cm (±51.06), which also was not statistically significant (*t* = 0.72, *p* = 0.480). The mean change was –10.60 cm (±65.73), and the effect size was –0.16, suggesting only a minimal effect.

As shown in Table 11 and Figure 10, these findings suggest that the digital game-based physical activity program used in this study may not have been sufficient to produce measurable improvements in explosive power within a short-term intervention period.

In the gender-specific subgroup analysis within the experimental group, male students showed virtually no change in explosive power, with a pre-test mean of 152.78 cm (±30.15) and a post-test mean of 152.83 cm (±31.02) (*p* = 0.999). Female students showed a slight decline, from 142.18 cm (±25.80) to 140.00 cm (±21.48), but the change was not statistically significant (*p* = 0.414). As shown in Table 12 and Figure 11, these findings indicate that the program did not have a notable effect on improving explosive power in either gender subgroup.

## 4. Discussion

This study aimed to examine the effects of a digital game-based physical activity program (AI-FIT) on health-related physical fitness indicators in elementary school students, including flexibility, muscular endurance, overall physical fitness (PAPS grade), cardiorespiratory endurance, and explosive power. The main findings are discussed as follows.

### 4.1. Changes in Flexibility

The AI-FIT program also proved to be an effective intervention for enhancing flexibility. These changes are not merely attributable to the program’s gamified appeal but rather to its structured inclusion of lower-body and core-centered stretching exercises (e.g., toe touches, ground touches, and planks), which were delivered with a specific frequency and number of sets and progressively intensified based on AI-monitored performance data.

This intervention model differs significantly from conventional physical education classes by providing a systematic framework for repetition and individualization in flexibility training. Particularly, flexibility tends to improve more rapidly than muscular or cardiorespiratory endurance, making it a suitable focus in early stages of intervention to promote self-efficacy [30]. Ahmed Khan, Ansari, and Azeemi [31] also found that improvements in flexibility serve as a psychological facilitator by boosting children’s self-confidence and sustained motivation for physical activity.

Furthermore, considering developmental characteristics of children, the upper elementary years (grades 4 to 6) may represent an optimal period for targeted flexibility interventions. Buhari and Joseph [32] reported that a gymnastics-based intervention for children aged 9–12 significantly improved flexibility in both boys and girls, with no statistically significant gender differences. This supports the findings of the present study, in which the AI-FIT program effectively enhanced flexibility regardless of gender.

Similarly, Comeras-Chueca et al. [33], in a systematic review, reported that active video games positively influence movement-related functionality, such as flexibility, in children and adolescents with healthy body weight. Staiano and Calvert [34] also emphasized that immersive game-based activities promote exercise adherence and repetition, contributing to physical fitness improvements. The present study provides empirical support for these findings by demonstrating that AI-driven digital interventions can improve fundamental fitness components such as flexibility.

Additionally, the consistent improvement observed across gender subgroups in this study suggests that the AI-FIT program functioned equitably, without favoring specific populations, and adapted effectively to a range of physical conditions. This reinforces its potential as a gender-inclusive intervention model for future fitness education programs.

### 4.2. Changes in Muscular Endurance

The experimental group participating in the AI-FIT program showed a statistically significant improvement in muscular endurance, as measured by the number of sit-ups performed in one minute. The effect size was large (Cohen’s *d* = 0.80), indicating that the digital game-based intervention effectively enhanced muscular endurance by promoting repeated physical activity among children. In particular, AI-FIT included multiple sets of bodyweight exercises focused on the upper body and core—such as arm walking, planks, mountain climbers, box runs, and burpees—which are well-suited to improving muscular endurance in elementary students, who are generally unfamiliar with resistance-based stimuli.

These findings are consistent with previous studies on resistance training. Faigenbaum et al. [35] demonstrated that high-repetition, moderate-intensity resistance training significantly improved muscular endurance in elementary school children. Similarly, a meta-analysis by Behringer et al. [36] confirmed that strength training programs for youth can be increasingly effective depending on the level of repetition and duration of intervention. Furthermore, Resaland et al. [37] also validated the potential of a school-based physical activity intervention (ASK) to improve muscular endurance through a cluster randomized controlled trial. These studies collectively support the notion that structured exercise programs contribute to fitness development and habit formation in children.

However, in the gender-specific subgroup analysis, both male and female students showed either slight decreases or minimal changes in muscular endurance, and the results did not reach statistical significance. While reduced statistical explosive power due to smaller sample sizes may partially explain this, a more plausible interpretation lies in the program’s intensity structure and emphasis across fitness components. Although the AI-FIT intervention included muscular endurance exercises such as planks and arm walking, the overall program design emphasized aerobic and coordination-based movements (e.g., shuttle runs, speed steps, and box runs) with shorter muscle-loading durations (e.g., 20 s per set), which may have limited the stimulus necessary to elicit significant adaptations in muscular endurance, particularly at the subgroup level. Additionally, progressive overload principles were applied across sessions via performance-based adjustments rather than structured resistance progression, which may not have been sufficient to induce statistically detectable improvements in muscular endurance.

Additionally, the possibility of a ceiling effect in subgroups with higher initial performance (e.g., male students) or limited improvement due to low baseline endurance and slower adaptation among female students should be considered.

### 4.3. Changes in Overall Physical Fitness (PAPS Grade)

The AI-FIT program demonstrated a positive effect on overall physical fitness in children. The experimental group showed significant improvements in the PAPS grade, a composite indicator of health-related fitness. This suggests that the program not only improved individual fitness components but also contributed to an overall enhancement of physical health. By incorporating various fitness elements—muscular endurance, flexibility, explosive power, and cardiorespiratory endurance—and adjusting training intensity through an AI-based feedback system, the program delivered structured and engaging training experiences tailored to each participant’s performance level.

Game-based physical activity programs have also been shown to promote not only fitness improvement but also emotional engagement, thereby enhancing children’s participation and adherence. According to a recent systematic review by Mo et al. [38], game-based physical education programs significantly enhance enjoyment among children and adolescents, which serves as a key driver of sustained physical activity participation and self-determined motivation. Rather than focusing solely on repetitive task completion, structured physical activities grounded in play offer children a sense of psychological satisfaction and success, thereby fostering more positive attitudes toward physical exercise.

In contrast, conventional repetition-focused physical education may neglect these emotional dimensions. Overly disciplined or performance-driven approaches can undermine children’s motivation for physical activity. Gu and Zhang [39] warned that children’s motivation for physical education tends to decline as grade level increases, which may adversely affect their willingness to engage in long-term physical activity.

To address the limitations of using ordinal scale data, this study supplemented analysis with non-parametric testing (Wilcoxon signed-rank test), thereby improving the reliability of the findings. Furthermore, gender-specific subgroup analyses showed significant improvements in overall physical fitness for both boys and girls, suggesting that the AI-FIT program is a universally effective instructional model, regardless of gender.

### 4.4. Changes in Cardiorespiratory Endurance

In this study, the experimental group did not show a statistically significant change in cardiorespiratory endurance, as measured by PEI, following participation in the AI-FIT program (*t* = 0.52, *p* = 0.607), with a negligible effect size (Cohen’s *d* = –0.12). In contrast, the control group experienced a significant decline in PEI scores over the same period (*t* = 3.36, *p* = 0.002), with a medium-to-large effect size (*d* = –0.56). These findings suggest that although the AI-FIT program may not have directly enhanced cardiorespiratory endurance, it may have played a protective role in buffering against fitness deterioration.

Previous studies on the effects of AVGs on cardiorespiratory endurance have yielded mixed results. According to a meta-analysis by Gao et al. [35], AVGs were found to be less effective than traditional exercise programs in improving cardiorespiratory endurance, potentially due to insufficient exercise intensity or duration. Similarly, Comeras-Chueca et al. [33] suggested that interventions aiming to improve cardiorespiratory endurance through AVGs should last at least 18 weeks to achieve meaningful outcomes. Given that the AI-FIT program in this study was implemented over 12 weeks, the duration alone may not fully explain the limited gains in cardiorespiratory endurance. A more likely explanation is that the intermittent nature of the digital game-based tasks—with short sets (20 s) and rest intervals (up to 1 min)—may not have provided sufficient continuous cardiovascular stimulus to induce aerobic adaptations, despite the overall program duration.

The gender-specific subgroup analysis also revealed no statistically significant improvements in cardiorespiratory endurance for either boys or girls, indicating that AVGs do not appear to produce differential effects by gender in this domain among children.

### 4.5. Changes in Explosive Power

In this study, no statistically significant changes were observed in explosive power—as measured by standing long jump performance—in either the experimental or control group. This suggests that short-term digital game-based physical activity interventions may have limited effects on improving explosive power.

According to a meta-analysis by Gao et al. [40], while AVGs have shown positive effects on children’s flexibility and cardiorespiratory endurance, their impact on high-intensity strength components such as explosive power appears limited.

Lloyd and Oliver [41] emphasized that the development of power and muscular strength in children and adolescents requires not only progressive overload and repeated training but also targeted technical instruction. This implies that digital exercise programs alone may not provide the sufficient training stimuli or biomechanical guidance necessary for developing explosive power.

Although explosive power is not typically categorized under health-related physical fitness, it was included in this study in accordance with the PAPS national fitness assessment framework. This decision reflects the importance of aligning research design with Korea’s educational standards for school-based physical fitness.

This may be attributed to multiple factors, including insufficient exercise intensity, lack of repetition, and the absence of technical instruction within the program design. To enhance explosive power outcomes, future interventions should incorporate high-intensity training elements, skill-based instruction, and possibly extend the duration of the program. Moreover, including tasks specifically designed to target explosive power development would strengthen the effectiveness of such interventions.

### 4.6. Limitation, Future Directions, and Practical Applications

Despite the positive implications of this study, several limitations must be acknowledged. First, this study employed a relatively small sample of 40 students in grades 4 to 6 from a single elementary school located in Cheonan, Chungcheongnam-do, South Korea. Although propensity score matching was used to ensure group homogeneity, the generalizability of the findings is limited. Larger-scale follow-up studies encompassing diverse regions, age groups, and cultural contexts are needed to validate these results [26,38].

Second, this study did not systematically control for external variables that may influence exercise outcomes, such as dietary habits, sleep duration, and daily physical activity levels. Future research should rigorously measure and control for these factors to more precisely isolate the pure effects of digital physical activity interventions.

Third, this study did not include an in-depth analysis of qualitative factors such as students’ exercise immersion, motivation, or perceptions of the program. This limits understanding of the underlying mechanisms driving program effectiveness. Future studies should adopt qualitative or mixed-methods approaches incorporating interviews, observations, and self-reports to explore participants’ subjective experiences.

Fourth, this study did not sufficiently account for structural changes in students’ lifestyles post-COVID-19, such as reduced physical activity levels and increased online learning. Future research should consider these socio-environmental factors and examine how shifts in daily routines after the pandemic may influence the relationship between physical activity and health-related fitness.

In light of these limitations, future research should (1) recruit more diverse and representative samples to enhance the generalizability of findings; (2) adopt more sophisticated study designs that control for external variables such as nutrition, sleep, and baseline activity levels; (3) incorporate psychological and social factors—such as exercise immersion, self-efficacy, and peer interactions—through qualitative or mixed-methods research; and (4) empirically investigate how digital exercise interventions function within the altered educational and lifestyle environments of the post-pandemic era.

Finally, although this study provides initial empirical support for the effectiveness of a digital game-based physical activity program, future research should expand to include comparative studies across different digital exercise platforms, long-term follow-up studies, and design-based research aimed at refining and improving such interventions.

This study also offers practical implications for elementary school teachers, curriculum designers, and EdTech developers seeking to enhance health-related physical fitness in children through personalized and engaging digital programs. The results contribute to theoretical discourse by demonstrating the potential of AI-integrated, adaptive physical education tools aligned with the TPACK framework. Ultimately, this study lays the groundwork for future innovations in personalized fitness education and equitable access to technology-enhanced physical activity in school settings.

## 5. Conclusions

This study examined the effects of the AI-FIT digital game-based physical activity program on the health-related fitness of elementary school students using a quasi-experimental design with propensity score matching. The results revealed significant improvements in flexibility, muscular endurance, and overall physical fitness (PAPS grade), with medium-to-large effect sizes. These findings suggest that AI-FIT effectively enhances key components of health-related fitness through AI-driven personalization and structured game-based exercises.

However, no significant gains were observed in cardiorespiratory endurance or explosive power, indicating the need for more targeted, high-intensity stimuli to address these components. While the short duration and limited sample size restrict generalizability, the program demonstrated practical feasibility and educational potential.

This study contributes to the growing body of evidence supporting the use of AI and EdTech in physical education. It provides foundational insights for developing future-oriented, personalized, and engaging physical activity interventions. Future research should involve larger, more diverse samples, long-term follow-up, and refinement of task-specific design to optimize fitness outcomes across all domains.

## Figures and Tables

**Figure 1 healthcare-13-01327-f001:**
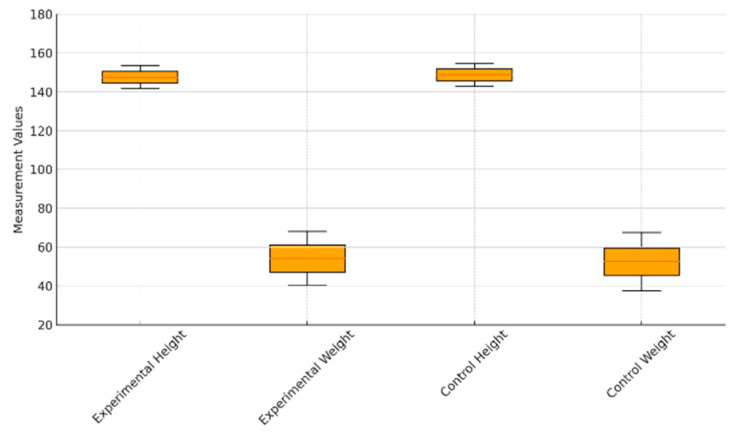
Comparison of body height and body weight between experimental and control groups.

**Figure 2 healthcare-13-01327-f002:**
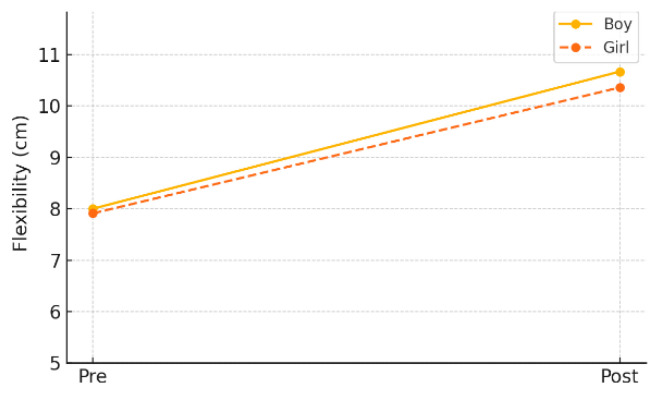
Pre–post changes in flexibility by gender (experimental group).

**Figure 3 healthcare-13-01327-f003:**
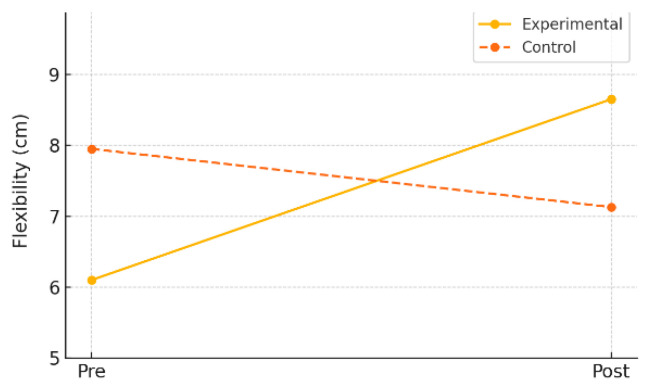
Pre–post changes in flexibility by group.

**Figure 4 healthcare-13-01327-f004:**
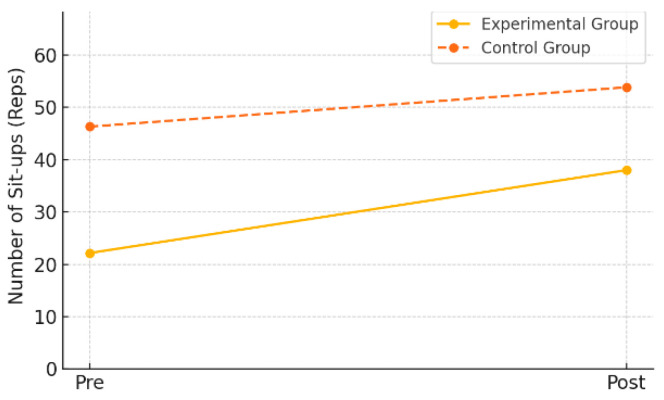
Pre–post changes in muscular endurance by group.

**Figure 5 healthcare-13-01327-f005:**
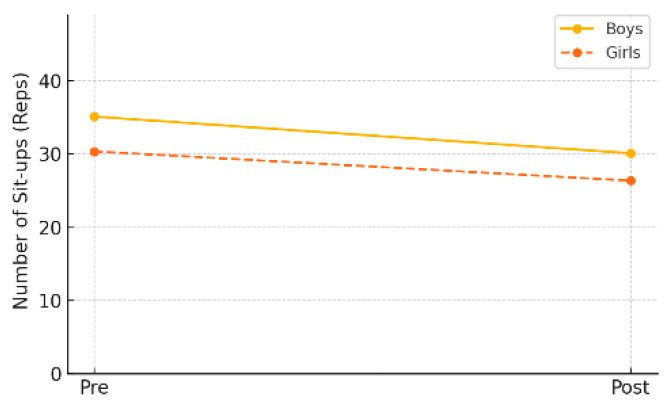
Pre–post changes in muscular endurance by gender (experimental group).

**Figure 6 healthcare-13-01327-f006:**
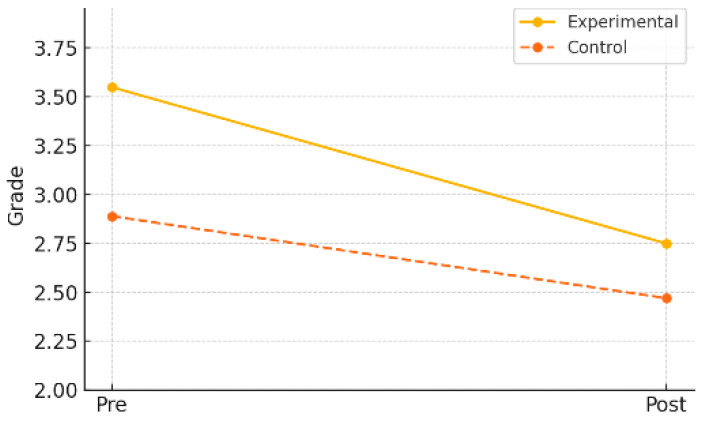
Pre–post changes in PAPS grade by group.

**Figure 7 healthcare-13-01327-f007:**
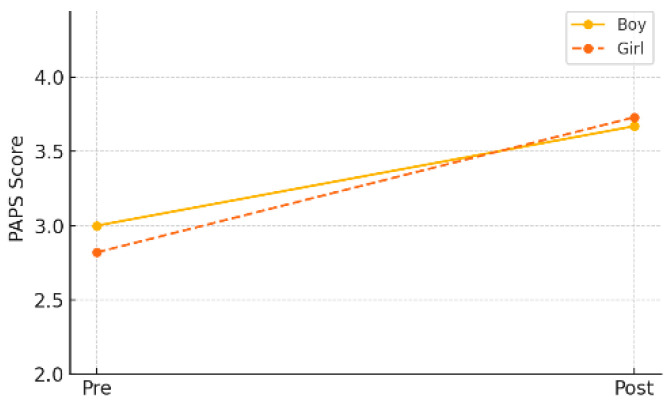
Pre–post changes in PAPS grade by gender (experimental group).

**Figure 8 healthcare-13-01327-f008:**
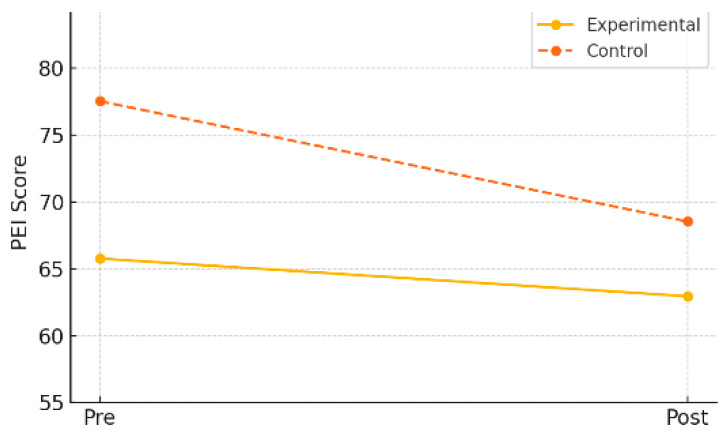
Pre–post changes in cardiorespiratory endurance by group.

**Figure 9 healthcare-13-01327-f009:**
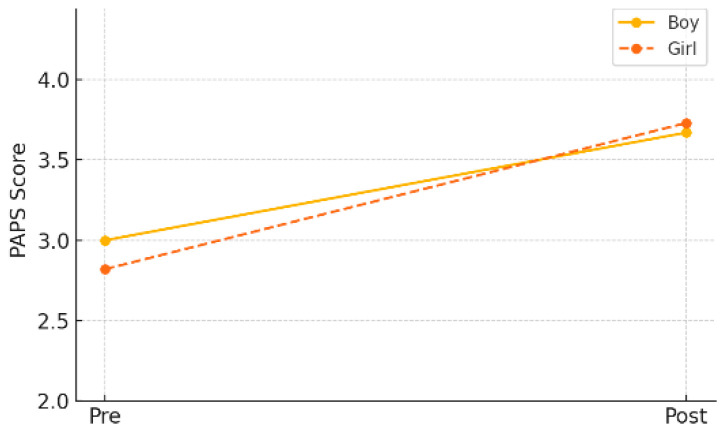
Pre–post changes in cardiorespiratory endurance by gender (experimental group).

**Figure 10 healthcare-13-01327-f010:**
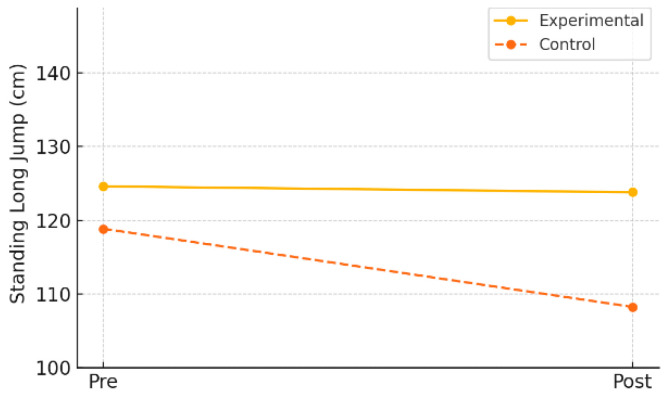
Pre–post changes in explosive power by group.

**Figure 11 healthcare-13-01327-f011:**
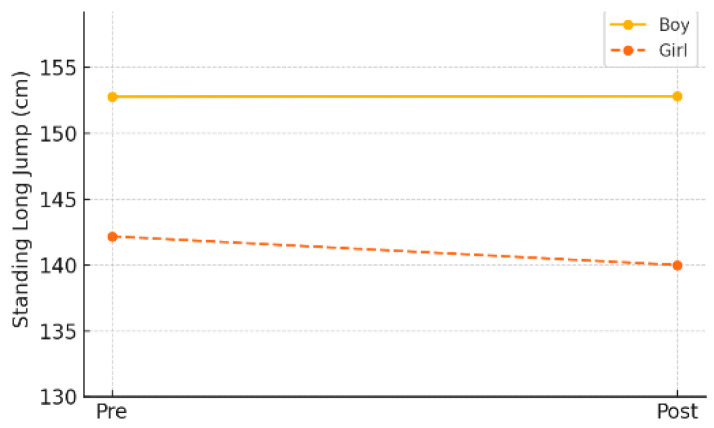
Pre–post changes in explosive power by gender (experimental group).

**Table 1 healthcare-13-01327-t001:** General characteristics of the participants.

Variable	Experimental Group (n = 20)	Control Group (n = 20)	Test Statistic	*p*-Value
Gender (M/F)	9/11	7/13	χ^2^(1) = 0.45	0.502
Grade (4/5/6)	6/11/3	8/9/3	χ^2^(2) = 0.21	0.765
Body height (cm)	147.6 ± 6.1	148.8 ± 5.9	t(40) = −0.59	0.560
Body weight (kg)	54.2 ± 14.1	52.6 ± 15.7	t(40) = 0.31	0.758

**Table 2 healthcare-13-01327-t002:** Exercise composition by fitness component in the AI-FIT program.

Fitness Component	Primary Exercises	Frequency per Week	Number of Sets	Duration per Set	Rest Interval
Muscular endurance	Arm walking, plank, mountain climber, shuttle run (box run), burpee	3 times	3 sets	20 s	1 min
Cardiorespiratory endurance	Speed step, shuttle run (box run), mountain climber	3 times	4 sets	20 s	1 min
Flexibility	Toe touch, ground touch, plank	2 times	2 sets	20 s	30 s
Explosive power	Standing long jump, burpee	2–3 times	3 sets	20 s	1–2 min

**Table 3 healthcare-13-01327-t003:** Pre–post comparison of flexibility changes between the experimental and control groups following participation in the AI-FIT program.

Variable	Group	Pre_Mean_Exp (SD)	Post_Mean_Exp (SD)	Δ(Post–Pre)	*t*	*p*-value	Cohen’s *d*
Flexibility (cm)	Experimental	6.10 ± 6.33	8.65 ± 5.89	+2.55 ± 2.83	−4.03	0.001 ***	0.90
	Control	7.95 ± 7.83	7.13 ± 8.82	−0.82 ± 5.23	0.70	0.491	−0.16

Δ = post–pre, SD = standard deviation, *d* = Cohen’s effect size, *p* < 0.001 (***).

**Table 4 healthcare-13-01327-t004:** Pre–post comparison of flexibility between male and female subgroups within the experimental group.

Variable	Boy		*p*-Value	Girl		*p*-Value
	Pre_Mean_Boy (SD)	Post_Mean_Boy (SD)		Pre_Mean_Girl (SD)	Post_Mean_Girl (SD)	
Flexibility (cm)	8.00 ± 5.21	10.67 ± 4.97	0.019 *	7.91 ± 5.81	10.36 ± 6.38	0.003 **
*t, d*	*t* = 1.17, *d* = 0.37			*t* = 0.90, *d* = 0.28		

SD = standard deviation, *t* = paired *t*-test, *d* = Cohen’s effect size, *p* < 0.05 (*), *p* < 0.01 (**).

**Table 5 healthcare-13-01327-t005:** Pre–post comparison of muscular endurance changes between the experimental and control groups following participation in the AI-FIT program.

Variable	Group	Pre_Mean_Exp (SD)	Post_Mean_Exp (SD)	Δ(Post–Pre)	*t*	*p*-Value	Cohen’s *d*
Muscular endurance (rep)	Experimental	22.15 ± 11.95	38.00 ± 26.49	+15.85 ± 19.78	−3.58	0.002 **	0.80
	Control	46.32 ± 36.03	53.84 ± 60.81	+7.53 ± 60.98	−0.54	0.597	0.12

Δ = post–pre, SD = standard deviation, *d* = Cohen’s effect size, *p* < 0.01 (**).

**Table 6 healthcare-13-01327-t006:** Pre–post comparison of muscular endurance between male and female subgroups within the experimental group.

Variable	Boy		*p*-Value	Girl		*p*-Value
	Pre_Mean_Boy (SD)	Post_Mean_Boy (SD)		Pre_Mean_Girl (SD)	Post_Mean_Girl (SD)	
Muscular endurance	35.11 ± 16.62	30.11 ± 15.35	0.083	30.36 ± 13.32	26.36 ± 13.63	0.058
*t, d*	*t* = −0.70, *d* = −0.22			*t* = −0.66, *d* = −0.21		

SD = standard deviation, *t* = paired *t*-test, *d* = Cohen’s effect size.

**Table 7 healthcare-13-01327-t007:** Pre–post comparison of overall physical fitness (PAPS grade) between the experimental and control groups following participation in the AI-FIT program.

Variable	Group	Pre_Mean_Exp (SD)	Post_Mean_Exp (SD)	Δ(Post–Pre)	*t*	*p*-Value	Cohen’s *d*
PAPS Score	Experimental	3.55 ± 0.60	2.75 ± 0.91	−0.80 ± 1.15	3.11	0.006 **	−0.69
	Control	2.89 ± 0.74	2.47 ± 1.17	−0.42 ± 0.77	2.39	0.028 *	−0.55

Δ = post–pre, SD = standard deviation, *d* = Cohen’s effect size, *p* < 0.05 (*), *p* < 0.01 (**).

**Table 8 healthcare-13-01327-t008:** Pre–post comparison of overall physical fitness (PAPS grade) between male and female subgroups within the experimental group.

Variable	Boy		*p*-Value	Girl		*p*-Value
	Pre_Mean_Boy (SD)	Post_Mean_Boy (SD)		Pre_Mean_Girl (SD)	Post_Mean_Girl (SD)	
PAPS Score	3.00 ± 0.87	3.67 ± 0.87	0.013 *	2.82 ± 0.70	3.73 ± 0.74	0.001 ***
*t, d*	*t* = 1.72, *d* = 0.54			*t* = 2.83, *d* = 0.89		

SD = standard deviation, *t* = paired *t*-test, *d* = Cohen’s effect size, *p* < 0.05 (*), *p* < 0.001 (***).

**Table 9 healthcare-13-01327-t009:** Pre–post comparison of cardiorespiratory endurance between the experimental and control groups following participation in the AI-FIT program.

Variable	Group	Pre_Mean_Exp (SD)	Post_Mean_Exp (SD)	Δ(Post–Pre)	*t*	*p*-Value	Cohen’s *d*
Endurance (PEI)	Experimental	65.77 ± 21.98	62.95 ± 18.88	−2.82 ± 23.54	0.52	0.607	−0.12
	Control	77.54 ± 17.28	68.54 ± 36.03	−9.00 ± 30.50	1.22	0.241	−0.30

Δ = post–pre, SD = standard deviation, *d* = Cohen’s effect size.

**Table 10 healthcare-13-01327-t010:** Pre–post comparison of cardiorespiratory endurance between male and female subgroups within the experimental group.

Variable	Boy		*p*-Value	Girl		*p*-Value
	Pre_Mean_Boy (SD)	Post_Mean_Boy (SD)		Pre_Mean_Girl (SD)	Post_Mean_Girl (SD)	
Endurance (count)	30.33 ± 11.42	31.11 ± 13.18	0.683	28.45 ± 15.13	33.00 ± 13.98	0.091
*t, d*	*t* = 0.14, *d* = 0.04			*t* = 0.70, *d* = 0.22		

SD = standard deviation, *t* = paired *t*-test, *d* = Cohen’s effect size.

**Table 11 healthcare-13-01327-t011:** Pre–post comparison of explosive power between the experimental and control groups following participation in the AI-FIT program.

Variable	Group	Pre_Mean_Exp (SD)	Post_Mean_Exp (SD)	Δ(Post–Pre)	*t*	*p*-Value	Cohen’s *d*
Explosive power (cm)	Experimental	124.60 ± 25.30	123.80 ± 47.08	−0.80 ± 47.76	0.07	0.941	−0.02
	Control	118.85 ± 34.10	108.25 ± 51.06	−10.60 ± 65.73	0.72	0.480	−0.16

Δ = post–pre, SD = standard deviation, *d* = Cohen’s effect size.

**Table 12 healthcare-13-01327-t012:** Pre–post comparison of explosive power between male and female subgroups within the experimental group.

Variable	Boy		*p*-Value	Girl		*p*-Value
	Pre_Mean_Boy (SD)	Post_Mean_Boy (SD)		Pre_Mean_Girl (SD)	Post_Mean_Girl (SD)	
Explosive power (cm)	152.78 ± 30.15	152.83 ± 31.02	0.999	142.18 ± 25.80	140.00 ± 21.48	0.414
*t, d*	*t* = 0.00, *d* = 0.00			*t* = −0.21, *d* = −0.06		

SD = standard deviation, *t* = paired *t*-test, *d* = Cohen’s effect size.

## Data Availability

The data presented in this study are available upon request from the corresponding author. The data are not publicly available because of the protection of personal information.

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
