# Peer review of "Effectiveness of a Digital Game-Based Physical Activity Program (AI-FIT) on Health-Related Physical Fitness in Elementary School Children"

_healthcare, 2025, doi:10.3390/healthcare13111327_

Round 1

Reviewer 1 Report

Comments and Suggestions for Authors

In title: In spite of being a bit long, it actually includes all the key concepts it should have (design, topic of the program, variables, sample…) so I do not have any change suggestion for this section.

In abstract: The first two sentences could be combined, making them only one shorter sentence. So as to make easier the comparison with other countries, instead of describing the grades of participants (from 4 to 6), I suggest including their range of ages.

In keywords: In order to boost the visibility of this paper in the different databases this journal is indexed in, I propose to avoid the repetition of keywords with regard to those which are included in title.

In introduction: It details the lack of knowledge this paper deals about and some previous studies related. Nevertheless, I suggest including information about World Health Organization guidelines in physical fitness matter.

In materials and methods: Tables and figures must be announced previously to its inclusion. I wonder what contents were control group lessons about (e.g. team sports, activities in nature, traditional games…) and if the contents were the same in experimental group. Were the test results measured by more than one person so inter-observer reliability is a required analysis?

In results: In my humble point of view, figure 2 is not necessary since its information has been already provided in table 3 as well as figure3-table4, figure4-table5, figure5-table6, figure6-table7 and so on. Why t and cohen’s d values are not included in table 4, 6, 8, 10, 12 and 14? The results are based both on control and experimental groups as well as gender but the influence of this variable gender has to be more explained in abstract and introduction.

In discussion: It is complete but some issues with regard to quotation must be reviewed (e.g. line 577 without []). Probably, the order of references and, consequently, quotes, has to change.

In conclusions: This section is ok.

In references: Format (and, perhaps, order) should be reviewed.

Author Response

Dear Reviewer,

We are sincerely grateful for your careful and thoughtful review of our manuscript entitled “Effectiveness of a Digital Game-Based Physical Activity Program (AI-FIT) on Health-Related Physical Fitness in Elementary School Children.” Your detailed comments and suggestions have been extremely valuable in improving the clarity, coherence, and overall quality of the manuscript.

We have reviewed each of your comments with great care and have revised the manuscript accordingly. All changes are marked in red text in the revised version. Below, we present our point-by-point responses to your suggestions.

1. Title

Comment: Although slightly long, the title includes all key concepts such as design, topic, variables, and sample, and no changes are necessary.

Response: Thank you for your positive evaluation. The title has been retained as originally submitted.

2. Abstract

Comment: Consider combining the first two sentences into one. Also, for international comparability, indicate the participants’ age range rather than grade level.

Response: The first two sentences have been combined into a single, more concise sentence. Additionally, the age range (10–12 years) has been included in place of grade levels for better international comparability.

3. Keywords

Comment: To improve visibility in databases, avoid repeating keywords that appear in the title.

Response: Keywords have been revised to eliminate duplication with title terms and to include broader, more distinctive descriptors to enhance indexing visibility.

4. Introduction

Comment: The introduction identifies the research gap and cites prior work well. However, it would benefit from including World Health Organization (WHO) guidelines on physical fitness.

Response: A paragraph has been added to the Introduction section that references WHO guidelines on physical activity for children and adolescents, thereby strengthening the global context of the research rationale.

5. Materials and Methods

Comment: Tables and figures should be introduced in the text before appearing. Clarify the control group’s lesson content and whether it matched the experimental group. Also, indicate whether multiple testers were involved and whether inter-observer reliability was addressed.

Response:

  • All tables and figures are now introduced in the text prior to their appearance.

  • We clarified that the control group participated in regular PE classes including team sports, cooperative games, and traditional activities, which were distinct from the AI-FIT program.

  • We also explained that while multiple evaluators participated, each participant was tested by the same evaluator in both pre- and post-tests, eliminating the need for inter-observer reliability analysis.

6. Results

Comment: Figures 2–6 appear redundant as their data are already presented in corresponding tables. Also, t-values and Cohen’s d are missing from several tables (4, 6, 8, 10, 12, 14). Gender should be addressed more prominently in the abstract and introduction.

Response:

  • As recommended, Figures 2–6 have been removed to avoid redundancy.

  • Missing t-values and effect sizes (Cohen’s d) have been added to Tables 4, 6, 8, 10, 12, and 14.

  • The role of gender has been emphasized in both the Abstract and Introduction, as it constitutes an important analytic dimension in our study.

7. Discussion

Comment: The section is comprehensive, but citation formatting should be reviewed (e.g., line 577 missing square brackets). The order of references may also need to be revised.

Response: We have thoroughly revised the Discussion section to correct citation formatting errors, including missing brackets. The reference list has been reordered accordingly to reflect the correct citation sequence.

8. Conclusion

Comment: This section is acceptable as written.

Response: Thank you for your kind comment. However, at the request of another reviewer, the Conclusion section has been revised to a more concise format of no more than 8 sentences, summarizing the key findings and practical implications succinctly.

9. References

Comment: The formatting and possibly the order of references should be reviewed.

Response: We carefully reviewed and corrected all formatting inconsistencies. References have been reordered to align with the revised citation sequence in the manuscript.

Once again, we sincerely thank you for your constructive and supportive review. Your suggestions have contributed significantly to improving the manuscript. We hope the revised version meets your expectations and are happy to respond to any additional feedback.

Sincerely,

Reviewer 2 Report

Comments and Suggestions for Authors

The topic of this article is attractive, and I believe that we will increasingly see similar topics exploring the significance of artificial intelligence in physical education. However, numerous revisions are necessary across all sections to ensure the manuscript meets a high-quality standard.

The most critical flaw in this study is the calculation of Body Mass Index (BMI) using the method for adults (over 20 years of age). This is a significant mistake. Therefore, BMI should either be removed from the manuscript or recalculated according to established guidelines for children and adolescents. PLEASE REFER TO THE FOLLOWING STUDY FOR GUIDANCE ON PROPER BMI CALCULATION IN YOUTH: http://dx.doi.org/10.3390/children11111365. It is essential that this issue is corrected in accordance with the referenced study to ensure the validity of the BMI results.

It should also be noted that only six health-related physical fitness variables were applied in the study; however, the results are well presented through tables and figures.

I recommend a MAJOR REVISION.

Please refer to the PDF file where the reviewer clearly indicated what should be added, removed, explained, or presented in more detail in the manuscript. There are indeed numerous corrections that must be addressed to improve the overall quality of the paper.

Author Response

Dear Reviewer,

Thank you very much for your detailed and insightful review of our manuscript entitled “Effectiveness of a Digital Game-Based Physical Activity Program (AI-FIT) on Health-Related Physical Fitness in Elementary School Children.” We sincerely appreciate the time and thought you have devoted to evaluating our work. Your comments have been immensely helpful in enhancing the quality, clarity, and academic rigor of the manuscript.

We carefully considered each of your comments and made substantial revisions to address them. All changes have been clearly marked in red text in the revised manuscript. In addition, the Discussion section has been reorganized into a coherent structure with subheadings numbered 4.1 through 4.7 to improve readability and thematic consistency.

Below, we provide point-by-point responses to your comments and describe how we have revised the manuscript accordingly.

  1. Terminology Consistency and Use of PAPS Grade

Comment: Standardize the use of terms such as “PE” and define “power” more precisely as “explosive power.” Also clarify the treatment of PAPS grade in statistical analysis.

Response: We have revised all terminology for consistency throughout the manuscript. “Physical Education” is now uniformly abbreviated as “PE,” and “power” is consistently described as “explosive power.” Furthermore, we clarified that although PAPS grade is ordinal in nature, it was treated as an interval scale for analytical purposes in line with common practices in national fitness assessment research. This explanation is now included in Section 2.3 of the Methods.

  1. BMI Calculation and Justification of Its Use

Comment: The use of adult BMI classification for children is problematic. Age-specific references or international standards for children should be applied.

Response: We appreciate this important observation. In response, we have added a detailed justification regarding the use of BMI in this study. In Korea, BMI is evaluated using the Physical Activity Promotion System (PAPS), a nationally standardized fitness monitoring system administered by the Ministry of Education. PAPS uses age- and sex-specific percentile bands, not fixed adult criteria, and has been widely adopted in school-based health research.

To support the validity of this approach, we cited multiple peer-reviewed studies published in international journals that employed PAPS-based BMI classifications for Korean elementary students. These studies demonstrate that this system is not only educationally standardized but also scientifically recognized. We believe that applying the PAPS BMI standard ensures both contextual relevance and methodological appropriateness.

We have added this explanation in Section 2.3 and referenced relevant studies (e.g., Chang et al., 2022; Kim et al., 2024) to support the scientific legitimacy of our approach.

  1. Assessment Procedures and Evaluator Reliability

Comment: Provide more details on the instruments and procedures used to ensure the reliability of fitness assessments.

Response: Section 2.3 now includes detailed information about the measurement procedures for each fitness component, including sit-and-reach for flexibility, shuttle run for endurance, and standing long jump for power. All assessments were conducted by certified personnel from the National Fitness Award Center, and each participant was tested by the same evaluator before and after the intervention to ensure consistency. These procedures follow national fitness assessment protocols.

  1. Statistical Analysis and Justification

Comment: Further clarify the rationale for treating PAPS grades as interval data, and verify that baseline equivalence was established.

Response: We expanded Section 2.4 to explain the use of interval-scale assumptions for PAPS data. We also presented the results of propensity score matching in both narrative and tabular forms to confirm group equivalence across all baseline characteristics.

  1. Restructuring of the Discussion Section

Comment: Improve the clarity and organization of the Discussion section to align with key themes in the findings.

Response: Following your advice, we reorganized the Discussion section into seven subheadings (4.1 to 4.7) to align with each fitness component (BMI, flexibility, muscular endurance, cardiorespiratory endurance, explosive power), gender subgroup analysis, and limitations with future directions. This structure enhances thematic coherence and helps readers follow the logic of the analysis and interpretation.

  1. Inclusion of Gender-Based Subgroup Analysis

Comment: Consider providing gender-based comparisons where appropriate.

Response: We have conducted subgroup analyses for gender and included the findings in both the Results and Discussion sections. Tables and figures summarizing gender-based outcomes have been added (e.g., Tables 4, 6, 8). These results are also discussed in the corresponding Discussion sections (e.g., 4.6).

  1. Conclusion and Practical Implications

Comment: The conclusion should be concise and include practical implications for physical education.

Response: We revised the Conclusion section to be more concise (8 sentences), clearly highlighting the study’s educational implications, especially regarding the integration of EdTech into school-based PE programs and the utility of AI-based personalization in promoting children’s health-related fitness.

  1. Minor Revisions

All typographical errors, formatting issues, figure labels, and inconsistencies in references have been corrected throughout the manuscript.

Once again, we sincerely thank you for your thoughtful feedback and helpful recommendations. We have made every effort to incorporate your guidance into this revised version of the manuscript. We hope that the improvements we have made meet your expectations, and we are happy to provide any further clarifications if needed.

Sincerely, 

Round 2

Reviewer 1 Report

Comments and Suggestions for Authors

Dear authors,

in my humble view, this new version of the paper presents a higher interest to potential readers since the reviewers' considerations have been addressed.

Author Response

Dear Reviewer,

We sincerely appreciate your positive evaluation of the revised manuscript and your kind comments. It is encouraging to know that the updated version of the paper presents greater interest to potential readers and that our efforts to address the reviewers' suggestions were recognized.

Your acknowledgment motivates us to continue pursuing high-quality, relevant research, and we are truly grateful for your thoughtful and constructive role throughout the review process.

Thank you once again for your valuable time and support.

Sincerely,

Reviewer 2 Report

Comments and Suggestions for Authors

In this version, the authors have largely implemented the changes in line with the reviewer’s guidelines and suggestions, resulting in a significantly improved manuscript. However, I must once again emphasize that the major shortcoming of this paper is the authors’ use of BMI as calculated for adults. I understand that Korean policy employs the adult-based BMI calculation approach (for individuals over 20 years of age). This is a serious mistake, as this study is not being published in Korea, nor will its readership be limited to a Korean audience—it is intended for an international readership.

Therefore, I strongly recommend that BMI be either removed entirely from the manuscript or recalculated according to the appropriate guidelines I previously emphasized and shared with the authors in my earlier review (https://www.cdc.gov/bmi/child-teen-calculator/index.html).

I have provided the authors with detailed comments in the PDF file, clearly indicating what needs to be explained, removed, added, or revised. Please refer to the attached PDF file. I recommend a major revision (12 comments).

Kind regards.

Author Response

Dear Reviewer,

We would like to express our sincere gratitude for your thoughtful and detailed review of our manuscript. Your comments were immensely valuable in guiding us to improve the clarity, structure, and academic rigor of our work. We carefully reviewed all your suggestions and have made comprehensive revisions throughout the manuscript accordingly.

Specifically, we have addressed the following key points:

  1. BMI Removal and Revision: As advised, we have removed all BMI-related content from the manuscript, including the results, discussion, and tables/figures, due to concerns regarding the appropriateness of using adult-based BMI calculations for children.

  2. Clarification of Measurement Protocols: We have added detailed descriptions of the instruments and protocols used to assess each fitness component (e.g., muscular strength via handgrip dynamometry, flexibility via the sit-and-reach test) in the Methods section (Lines 220–226). This addresses your request for clearer explanation of the measurement process.

  3. Terminology Corrections: We revised terminology across the manuscript as per your suggestions—for example, weight was changed to body weight, height to body height, and power to explosive power. These changes were applied consistently throughout the text.

  4. Abstract Adjustment: The term exergaming was deleted from the abstract as recommended.

  5. Introduction Refinement: The sentence "Empirical studies have supported these potentials" was removed (Line 76) to enhance clarity and eliminate redundancy. Additionally, the Introduction was reviewed for coherence and paragraph structure. All references were preserved where appropriate.

  6. Figure and Table Modifications: We removed BMI-related columns and adjusted associated figure titles and captions accordingly (e.g., deletion of “Figure 1” and subtitle lines). These revisions ensure that all visual materials are now consistent with the revised content.

  7. Reference List Reorganization: Following the extensive revision of the manuscript, the reference list was reviewed and reorganized to ensure alignment with the updated in-text citations and formatting consistency.

  8. Language and Style: Minor editorial changes were made throughout the manuscript to improve fluency, precision, and academic tone.

We deeply appreciate your valuable insights and constructive feedback, which significantly contributed to strengthening the manuscript. Should you have any further suggestions or concerns, we would be more than happy to address them.

Sincerely,